# A chemical link between methylamine and methylene imine and implications for interstellar glycine formation

Prasad Ramesh Joshi[1] & Yuan-Pern Lee [1,2,3✉]

Methylamine $CH_3NH_2$ is considered to be an important precursor of interstellar amino acid because hydrogen abstraction might lead to the aminomethyl radical $\bullet CH_2NH_2$ that can react with $\bullet HOCO$ to form glycine, but direct evidence of the formation and spectral identification of $\bullet CH_2NH_2$ remains unreported. We performed the reaction $H + CH_3NH_2$ in solid $p$-$H_2$ at 3.2 K and observed IR spectra of $\bullet CH_2NH_2$ and $CH_2NH$ upon irradiation and when the matrix was maintained in darkness. Previously unidentified IR spectrum of $\bullet CH_2NH_2$ clearly indicates that $\bullet CH_2NH_2$ can be formed from the reaction $H + CH_3NH_2$ in dark interstellar clouds. The observed dual-cycle mechanism containing two consecutive H-abstraction and two H-addition steps chemically connects $CH_3NH_2$ and $CH_2NH$ in interstellar media and explains their quasi-equilibrium. Experiments on $CD_3NH_2$ produced $CD_2HNH_2$, in addition to $\bullet CD_2NH_2$ and $CD_2NH$, confirming the occurrence of H addition to $\bullet CD_2NH_2$.

[1] Department of Applied Chemistry and Institute of Molecular Science, National Yang Ming Chiao Tung University, Hsinchu, Taiwan. [2] Center for Emergent Functional Matter Science, National Yang Ming Chiao Tung University, Hsinchu, Taiwan. [3] Institute of Atomic and Molecular Sciences, Academia Sinica, Taipei, Taiwan. ✉email: yplee@nycu.edu.tw

For the origin of life on earth it has long been presumed that prebiotic molecules were delivered from interstellar space through meteorites or comets or asteroids[1]. The simplest amino acid, glycine $NH_2CH_2C(O)OH$, is a key building block of proteins. Both glycine and methylamine, $CH_3NH_2$, were detected in comets Wild 2 and 67P/Churyumov-Gerasimenko (67P/C-G); these observations provided strong evidence for a cosmic origin of amino acids on Earth[2,3], but the nature of the formation of glycine and other prebiotic molecules during the star-forming process remains unclear.

Several paths have been proposed for the formation of glycine according to theoretical calculations. Among them, the radical–radical reactions

$$\bullet CH_2NH_2 + \bullet HOCO \rightarrow NH_2CH_2C(O)OH \qquad (1)$$

$$\bullet NH_2 + \bullet CH_2C(O)OH \rightarrow NH_2CH_2C(O)OH \qquad (2)$$

appear to be more important than the ionic channels[4]. Garrod[5] and Suzuki et al[6]. employed a model consisting of processes in the gaseous phase, on grain surface, and in bulk ice in hot cores to indicate the key roles of reactions (1) and (2) in the formation of glycine on the grains. Sato et al. employed state-of-the-art DFT calculations and reported that reaction (1) is the most feasible route[7]; these authors proposed that the aminomethyl radical, $\bullet CH_2NH_2$, might be produced from successive hydrogenation of HCN or H abstraction from $CH_3NH_2$ by $\bullet OH$ or $\bullet NH_2$[5–9]. A similar theoretical chemical model involving $NH_3$ and $\bullet HOCO$ was also proposed to understand the main routes for the formation and decomposition of $CH_3NH_2$[10].

Laboratory investigations to produce glycine from smaller precursors mimicking interstellar conditions have been extensive. UV photolysis and electron bombardment of various interstellar ice analogues such as $CO/NH_3/H_2O$ or $H_2O/CH_3NH_2/CO_2$ at low temperatures were demonstrated to yield glycine[11–18]. Recently, Ioppolo et al.[19] observed glycine formation from ices containing $CH_3NH_2$, CO, $O_2$, and atomic H under conditions similar to dark interstellar clouds; this result indicates that glycine can be formed with no need for energetic irradiation such as UV photons or cosmic rays, that is, during a much earlier star-formation stage than previously assumed. These authors proposed that glycine was produced via a barrierless radical-radical surface reaction (1), in which $\bullet CH_2NH_2$ was produced through H abstraction from $CH_3NH_2$ by $\bullet OH$ (produced from $H + O_2$) or H atom and $\bullet HOCO$ was produced via reaction between $\bullet OH$ and CO.

Even though the radical $\bullet CH_2NH_2$ plays a key role in the formation of glycine, its spectrum and mechanism of formation have yet to be directly identified. Bossa et al. reported that the isolation of this radical is difficult because of the reformation of $CH_3NH_2$ after the recombination of $\bullet CH_2NH_2$ with H atom; they consequently employed CO as an H-atom scavenger to diminish the recombination, but observed only formamide and N-methylformamide, not $\bullet CH_2NH_2$, after VUV (vacuum ultraviolet) irradiation of a $CH_3NH_2/CO$ binary ice mixture[20].

Here, we present direct experimental evidence, via IR spectra, of the formation of $\bullet CH_2NH_2$ and $CH_2NH$ from the reaction of H atom with $CH_3NH_2$ via a H-abstraction tunneling reaction at low temperature, even in darkness. Furthermore, our experimental results showed a tight chemical connection between $CH_3NH_2$ and $CH_2NH$ through dual H-abstraction and H-addition cycles.

## Results and discussion

To perform H-atom reactions in the laboratory, we co-deposited $Cl_2$, $CH_3NH_2$, and para-hydrogen ($p$-$H_2$) at 3.2 K and irradiated the matrix with light at 365 nm from a light-emitting diode, followed by IR irradiation. The UV photolysis of $Cl_2$ at 365 nm

generated Cl atom, which is stable toward $H_2$ because the reaction $Cl + H_2 \rightarrow HCl + H$ is endothermic and has a large barrier[21]. The subsequent IR irradiation excites $H_2$ from $v = 0$ to $v = 1$ to overcome the energetic limitations so that the Cl atom reacts with $H_2$ ($v = 1$) to form $HCl + H$. The H atom thus produced reacted with $CH_3NH_2$ during IR irradiation, but the reaction continued even when the matrix was maintained in darkness for a long period because the H atom could migrate slowly in the matrix via tunneling reactions to break and form neighboring H−H bonds (so-called quantum diffusion) to approach $CH_3NH_2$ and react via tunneling reactions. The efficient production of Cl from photolysis of $Cl_2$ in a low-temperature matrix requires a diminished cage effect, the production of H requires $H_2$ ($v = 1$), and the migration of H in darkness requires quantum tunneling reaction; all these become possible only in quantum solid $p$-$H_2$ having associated unique properties[22,23].

**Observation of methylamine radical ($\bullet CH_2NH_2$) and methylene imine ($CH_2NH$).** The IR spectrum of a $CH_3NH_2/Cl_2/p$-$H_2$ (1/10/10000) matrix is shown in Supplementary Fig. 1a. The difference spectra after photolysis of the matrix at 365 nm for 30 min, subsequent IR irradiation for 90 min, maintenance of the matrix in darkness for 10 h, and secondary photolysis at 460 nm for 30 min are shown in Supplementary Fig. 1b–e, respectively. New features that appeared after IR irradiation, decreased by ~6% after being in darkness, and decreased by ~30% after secondary photolysis are indicated as group A, whereas those that appeared after IR irradiation, remained nearly constant after being in darkness, and increased by ~90% after secondary photolysis are indicated as group B. The difference spectrum after secondary photolysis at 460 nm (Supplementary Fig. 1e) in representative spectral regions is reproduced in Fig. 1b; lines in group A are pointing downward and those in group B are pointing upward, as indicated with color-coded arrows and labels.

The lines in group A at 3500.5, 3403.6, 3143.3, 3042.6, 1609.9, 1213.6, and 685.5 $cm^{-1}$ agree well, in terms of wavenumbers and relative intensities, with the IR stick spectrum simulated for $\bullet CH_2NH_2$ (Fig. 1a) according to the scaled harmonic vibrational wavenumbers and IR intensities predicted with the B3LYP/aug-cc-pVTZ method; the scaling method is discussed in the Methods

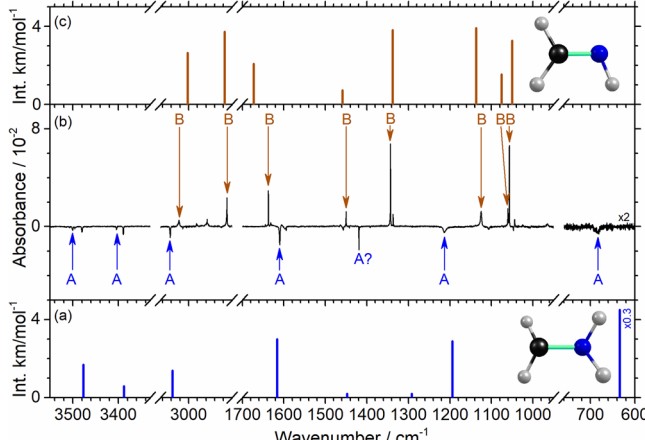

**Fig. 1 Comparison of observed lines in groups A ($\bullet CH_2NH_2$) and B ($CH_2NH$) with theoretical calculations. a** IR stick spectrum of aminomethyl radical $\bullet CH_2NH_2$. **b** IR difference spectrum after secondary photolysis at 460 nm of a UV/IR- irradiated $CH_3NH_2/Cl_2/p$-$H_2$ matrix after being maintained in darkness for 10 h; **c** IR stick spectrum of methylene imine $CH_2NH$. Both IR stick spectra in **a** and **c** were simulated according to scaled harmonic vibrational wavenumbers and IR intensities calculated with the B3LYP/aug-cc-pVTZ method.

**Table 1 Comparison of observed wavenumbers and relative IR intensities of •CH$_2$NH$_2$ in solid $p$-H$_2$ with their scaled harmonic vibrational wavenumbers and IR intensities predicted with the B3LYP/aug-cc-pVTZ method.**

| Mode | Sym. | $p$-H$_2$ | | B3LYP/aug-cc-pVTZ | | | $\Delta\nu$ [e] (cm$^{-1}$) |
|---|---|---|---|---|---|---|---|
| | | $\nu$/cm$^{-1}$ | Intensity[a] (%) | $\nu$[b] (cm$^{-1}$) | Intensity (km mol$^{-1}$) | Mode description[c] | |
| $\nu_1$ | A′ | 3403.6 | 20 | 3388 | 6 | $\nu_s$ NH$_2$ | −15.6 |
| $\nu_2$ | A′ | 3042.6 | 45 | 3037 | 14 | $\nu_s$ CH$_2$ | −5.6 |
| $\nu_3$ | A′ | 1609.9 | 100 | 1616 | 30 | $\rho$ NH$_2$ | 6.1 |
| $\nu_4$ | A′ | 1419.2 (?) | 5 | 1448 | 2 | $\delta$ CH$_2$ | — |
| $\nu_5$ | A′ | 1213.6 | 93 | 1194 | 29 | $\nu$ CN/$\rho$ CH$_2$ | −19.6 |
| $\nu_6$ | A′ | 685.5 | d | 634 | 151 | $\omega$ NH$_2$ | −51.5 |
| $\nu_7$ | A′ | — | — | 562 | 128 | $\omega$ CH$_2$ | — |
| $\nu_8$ | A″ | 3500.5 | 35 | 3477 | 17 | $\nu_a$ NH$_2$ | −23.5 |
| $\nu_9$ | A″ | 3143.3 | 11 | 3134 | 11 | $\nu_a$ CH$_2$ | −9.3 |
| $\nu_{10}$ | A″ | — | — | 1292 | 2 | $\gamma$ CH$_2$/$\gamma$ NH$_2$/$ip$-$def$ | — |
| $\nu_{11}$ | A″ | — | — | 913 | 1 | $\gamma$ NH$_2$/$\gamma$ CH$_2$ | — |
| $\nu_{12}$ | A″ | — | — | 432 | 27 | $\tau$ CH$_2$/$\tau$ NH$_2$ | — |

[a]Integrated intensity relative to the most intense line at 1609.9 cm$^{-1}$ ($\nu_3$). [b]Harmonic vibrational wavenumber scaled with the linear equations $y = (0.9810 \pm 0.0126) \, x - (2.9 \pm 16.9)$ and $y = (0.8907 \pm 0.0084) \, x + (233.0 \pm 27.2)$ for regions below and above 2500 cm$^{-1}$, respectively. [c]Approximate mode description; $\nu$: stretch, $\rho$: scissor, $\delta$: bend, $\gamma$: rock, $\omega$: wag, $def$: deformation, $\tau$: twist, $ip$: in-plane, subscript $a$: antisymmetric, and subscript $s$, symmetric. [d]Intense absorption of solid $p$-H$_2$ near 710 cm$^{-1}$ interfered with the intensity measurement. [e] Deviation $\Delta\nu = \nu_{calculated} - \nu_{experimental}$.

section. To understand the perturbations of H$_2$ on the IR spectrum of •CH$_2$NH$_2$, we performed calculations also on •CH$_2$NH$_2$ surrounded by eighteen H$_2$ molecules, either in a hexagonal-closed pack ($hcp$) lattice or randomly (free optimization). The resultant vibrational wavenumbers and IR intensities are compared in Supplementary Table 1 and the simulated IR stick spectra of •CH$_2$NH$_2$ are presented in Supplementary Fig. 2 to compare with calculations for gaseous •CH$_2$NH$_2$ and experiments. The perturbation by H$_2$ is small (with average absolute deviations 8.8 ± 6.0 and 14.8 ± 8.5 cm$^{-1}$ from the gaseous phase; listed errors represent one standard deviation in fitting) and within calculation errors. This is in line with the fact that observed IR spectra of matrix-isolated species typically showed <1% matrix shifts so that comparison of observed vibrational wavenumbers with predictions of gaseous species was generally performed. The observed lines in group A agree poorly with stick IR spectra of other possible products, as shown in Supplementary Fig. 3.

The symmetric and antisymmetric NH$_2$-stretching modes of •CH$_2$NH$_2$ predicted near 3388 and 3477 cm$^{-1}$ were observed at 3403.6 and 3500.5 cm$^{-1}$, respectively. The symmetric and antisymmetric CH$_2$-stretching modes predicted near 3037 and 3134 cm$^{-1}$ were observed at 3042.6 and 3143.3 cm$^{-1}$, respectively. The NH$_2$-scissoring mode predicted at 1616 cm$^{-1}$ agrees with the observed feature at 1609.9 cm$^{-1}$. The CN-stretching mode coupled with CH$_2$-scissoring, predicted near 1194 cm$^{-1}$, was observed at 1213.6 cm$^{-1}$. The most intense line predicted for the NH$_2$-wagging mode at 634 cm$^{-1}$ was observed at 685.5 cm$^{-1}$. Experiments and calculations are compared in Table 1. The observed features in group A can hence be clearly assigned to •CH$_2$NH$_2$. The average absolute deviation between experiment and prediction is 18.7 ± 15.2 cm$^{-1}$ (1.06 ± 0.8%) for •CH$_2$NH$_2$. The large deviation for $\nu_6$ (CH$_2$ wag) is typical for this mode because of the inadequacy in describing the double-well potential experienced by N atom, similar to NH$_3$. All lines of •CH$_2$NH$_2$ located in our detection spectral range with predicted IR intensity greater than 6 km mol$^{-1}$ were observed; predicted lines near 1448 and 1292 cm$^{-1}$ have intensity ~2 km mol$^{-1}$ too small to be observed.

The lines in group B at 3260.2, 3022.4, 2912.1, 1637.4, 1450.2, 1343.3, 1125.1, 1060.3, and 1056.9 cm$^{-1}$ are readily assigned to methylene imine CH$_2$NH, of which the IR spectrum in solid $p$-H$_2$

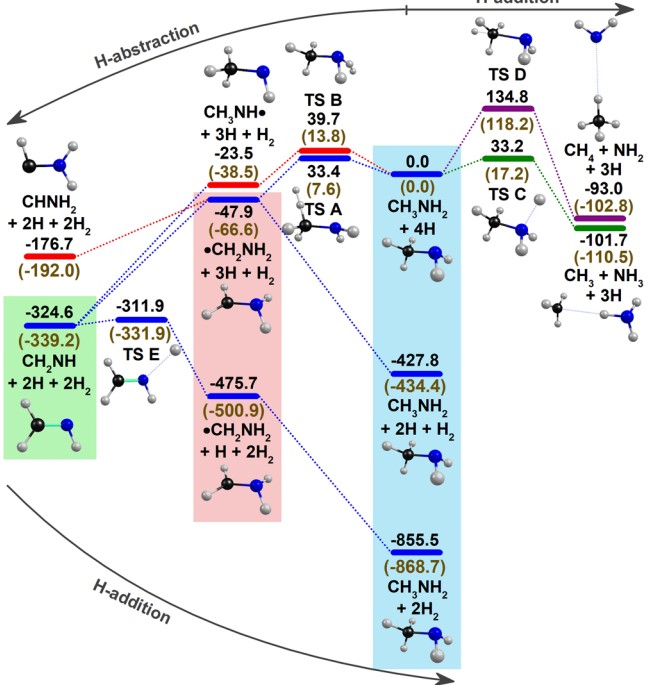

**Fig. 2 Potential-energy scheme of H-addition and H-abstraction of CH$_3$NH$_2$ calculated with the CCSD(T)/aug-cc-pVTZ//B3LYP/aug-cc-pVTZ method.** Energies corrected for zero-point vibrational energy (ZPVE) are in kJ mol$^{-1}$; those calculated with the B3LYP/aug-cc-pVTZ methods are listed in parentheses for comparison.

was recorded by Ruzi and Anderson on photodissociation at 193 nm of N–methylformamide in solid $p$-H$_2$[24]. The observed vibrational wavenumbers and relative intensities also agree with those predicted for CH$_2$NH, as shown in Fig. 1c. A comparison of experiments with theoretical calculations is presented in Supplementary Table 2.

To explore the possible products of reaction H + CH$_3$NH$_2$, we performed quantum-chemical calculations with the CCSD(T)/aug-cc-pVTZ//B3LYP/aug-cc-pVTZ method. The potential-energy scheme for H-abstraction (left side) and H-addition (right

side) is shown in Fig. 2; to be self-consistent in terms of energy and species involved, we included all H atoms and $H_2$ involved in this reaction network. The first H-abstraction from either moiety $CH_3$ or $NH_2$ of $CH_3NH_2$ (light blue background) results in the formation of $\cdot CH_2NH_2$ (pink background) or $CH_3NH\cdot$, respectively; abstraction on the $CH_3$ site has a smaller barrier and is more exothermic. The H-addition to either moiety $NH_2$ or $CH_3$ of $CH_3NH_2$ results in the rupture of the $C-N$ bond to form $CH_3 + NH_3$ (the most exothermic) or $CH_4 + NH_2$ (involving the largest barrier). Both radicals can proceed with further H-abstraction without barrier to form closed-shell methylene imine $CH_2NH$ (light green background). The H-addition to $\cdot CH_2NH_2$ to reproduce $CH_3NH_2$ is barrierless, and that to $CH_2NH$ to form $\cdot CH_2NH_2$ has a small barrier ($\sim 13$ kJ mol$^{-1}$).

The observation of $\cdot CH_2NH_2$ (group A) and $CH_2NH$ (group B) agrees with the predicted minimal paths for consecutive H-abstraction of $CH_3NH_2$. The H-abstraction of $\cdot CH_2NH_2$ to form $CH_2NH$ is barrierless, whereas that of $CH_3NH_2$ to form $\cdot CH_2NH_2$ has barrier $\sim 33$ kJ mol$^{-1}$, small enough for the tunneling reaction to occur even at 3.2 K. Our previous observations of H-abstraction of methanol[25], formamide[26], methyl formate[27], acetamide[28], acetic acid[29], and glycine[30] by H atoms showed predicted barriers of 36, 26, 41, 41, 43, and 29 kJ mol$^{-1}$, respectively. Although the calculations showed that H abstraction on the amino hydrogen to form $CH_3NH\cdot$ has a barrier $\sim 40$ kJ mol$^{-1}$, we did not observe $CH_3NH\cdot$, which can be readily identified at 3241.5, 2828.9, 2795.5, 1365.8, 1025.2 cm$^{-1}$ in solid $p$-$H_2$ as reported by Ruzi and Anderson[24].

The destruction of $\cdot CH_2NH_2$ to form $CH_2NH$ upon irradiation at 460 nm also agrees with the TD-B3LYP/aug-cc-pVTZ calculations showing an absorption band near 450 nm corresponding to HOMO ($\alpha$) $\rightarrow$ LUMO ($\alpha$) for $\cdot CH_2NH_2$ (Supplementary Figs. 4 and 5 and Supplementary Table 3), but no absorption for $CH_3NH_2$ at wavelength greater than 250 nm. We observed also the formation of a small proportion of $NH_3$ ($\sim 8\%$ of $\cdot CH_2NH_2$), indicating that H addition to $CH_3NH_2$ produced $CH_3 + NH_3$. We observed also $CH_3Cl$, likely produced from the reaction of $CH_3$ with Cl.

**Isotopic Substitution reaction**. We performed also experiments on partially deuterated methyl amine, $CD_3NH_2$. The representative spectra at various stages are depicted in Supplementary Fig. 6. The IR difference spectrum on secondary photolysis at 460 nm of the UV/IR-irradiated matrix after being maintained in darkness for 10 h is compared with IR stick spectra predicted for $\cdot CD_2NH_2$ and $CD_2NH$ in Supplementary Fig. 7. Lines in group A′ match with IR lines predicted for $\cdot CD_2NH_2$. Similarly, lines in group B′ match with those predicted for $CD_2NH$. In addition to lines in group A′ and group B′, we observed lines at 2125.2, 996.2, 937.6, 838.7, and 769.7 cm$^{-1}$, denoted group C′, that can be assigned to $CD_2HNH_2$, as presented in Supplementary Fig. 8. This observation confirms that H addition to $\cdot CD_2NH_2$ produced $CD_2HNH_2$. A comparison of experimental results and calculations for $\cdot CD_2NH_2$, $CD_2NH$, and $CD_2HNH_2$ is listed in Supplementary Tables 4−6. This isotopic experiment provides not only spectral confirmation of $\cdot CH_2NH_2$ and $CH_2NH$, but also direct evidence of the conversion of $\cdot CH_2NH_2$ back to $CH_3NH_2$ by H-addition, consistent with the report by Oba et al. that H−D substitution of solid $CH_3NH_2$ (and its isotopologues) is more rapid in the $CH_3$ moiety than in the $NH_2$ moiety when $CH_3NH_2$ reacts with H or D atoms under astrophysically relevant conditions[31]. Although we performed no experiment on $CD_3ND_2$ to confirm that H + $CD_2ND \rightarrow \cdot CD_2NDH$ occurred, we expect this reaction to occur because of a small barrier.

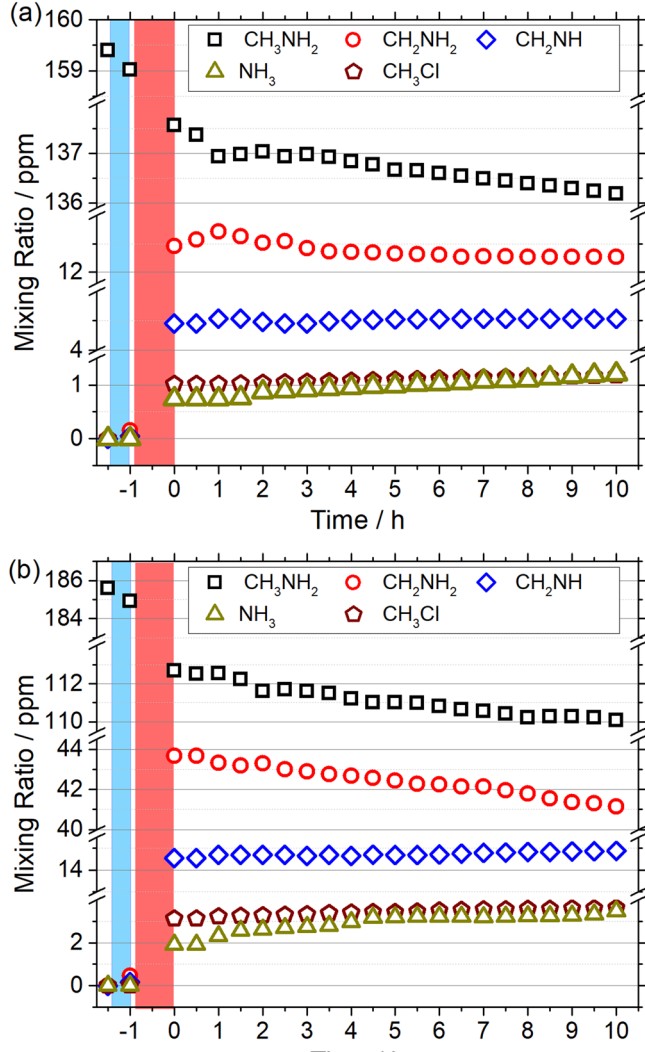

**Fig. 3 Temporal evolution of mixing ratios of $CH_3NH_2$, $\cdot CH_2NH_2$, $CH_2NH$, $NH_3$, and $CH_3Cl$ upon UV and IR irradiation of $CH_3NH_2/Cl_2/p$-$H_2$ matrices, followed by maintenance in darkness. a** [H]$_0$/[$CH_3NH_2$]$_0 \approx 2.1$ and [$CH_3NH_2$]$_0 = 159$ ppm, **b** [H]$_0$/[$CH_3NH_2$]$_0 \approx 7.2$ and [$CH_3NH_2$]$_0 = 186$ ppm. The regions shaded with blue and red correspond to periods of UV and IR irradiation, respectively.

**Temporal profiles and reaction mechanism**. The temporal evolution of the mixing ratios of each species is presented in Fig. 3 for two conditions with [H]$_0$/[$CH_3NH_2$]$_0 \approx 2.1$ ([$CH_3NH_2$] $\approx$ 159 ppm) and [H]$_0$/[$CH_3NH_2$]$_0 \approx 7.2$ ([$CH_3NH_2$]$_0 \approx$ 186 ppm); details are discussed in Supplementary Note 1. $\cdot CH_2NH_2$ was the dominant product, followed by $CH_2NH$. In the H-deficient experiment, we observed an initial increase of $\cdot CH_2NH_2$, followed by a decrease to reach a constant mixing ratio, consistent with the two-step mechanism for the formation from the first H abstraction of $CH_3NH_2$ and the destruction due to the second H abstraction; the *anti*-correlation between the mixing ratios of $CH_3NH_2$ and $\cdot CH_2NH_2$ was clearly visible. In the H-rich experiment, $\cdot CH_2NH_2$ and $CH_2NH$ increased significantly upon IR irradiation, and $\cdot CH_2NH_2$ continuously decreased in darkness, indicating more significant H abstraction of $\cdot CH_2NH_2$ due to the presence of more H atoms.

The observation of consecutive H abstraction of $CH_3NH_2$ to form $\cdot CH_2NH_2$ and $CH_2NH$, and their H-addition to reform

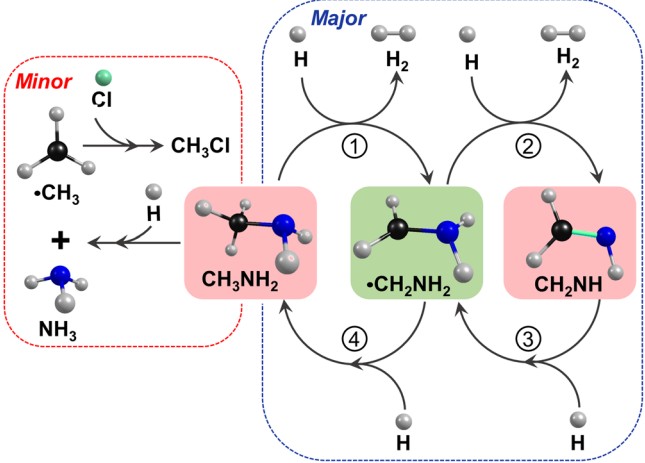

**Fig. 4 Dual-cycle mechanism of H-abstraction and H-addition reactions connecting $CH_3NH_2$, $\cdot CH_2NH_2$, and $CH_2NH$ and the formation of $CH_3$ and $NH_3$.** The area on the right with blue dotted boundary represents the major dual-cycle channels connecting $CH_3NH_2$, $\cdot CH_2NH_2$, and $CH_2NH$ with two sets of H-abstraction and H-addition reactions. The area on the left with red dotted boundary represents minor reactions channels of the decomposition of $CH_3NH_2$ to form $CH_3$ and $NH_3$ upon H addition).

$CH_3NH_2$ and $\cdot CH_2NH_2$, respectively, connects $CH_3NH_2$, $\cdot CH_2NH_2$, and $CH_2NH$ via a dual-cycle mechanism shown in Fig. 4, similar to that among formamide $HC(O)NH_2$, $H_2NCO$ and $HNCO$[26]. The first H-abstraction channel (reaction 1) depicted in Fig. 4 was reported by Garrod[5] and Suzuki[6] in their theoretical models. Hydrogenation of solid HCN and $CH_2NH$ at low temperature conducted by Theule et al.[32] resulted in the formation of $CH_3NH_2$ directly, indicating the presence of two consecutive H addition (reactions 3 and 4), even though $\cdot CH_2NH_2$ was not observed directly. In the present study, all guest molecules are well isolated in solid $p$-$H_2$ at low temperature so that free radicals such as $\cdot CH_2NH_2$ have much better chance to be trapped and maintained. Furthermore, because the hydrogenation experiments by Theule et al.[32] were carried out by hydrogen bombardment, $\cdot CH_2NH_2$ is expected to react readily with a second hydrogen atom to form the end product $CH_3NH_2$. Our observations of the formation of $\cdot CH_2NH_2$ and $CH_2NH$ in darkness and the formation of $CD_2HNH_2$ from H reactions with $CD_3NH_2$ further support the dual-cycle mechanism.

We understand that our experimental conditions do not mimic the ISM conditions closely, so our results cannot be applied directly to the reactions in the ISM. For example, in the case of water ice environments, the interaction between water and the guest species might be stronger so that the stability and reactivity of radicals are different from the gaseous phase, as demonstrated by the simulations of radical–radical reactions on icy surfaces by Enrique-Romero et al.[33] Nevertheless, our results clearly indicate that reaction of H with methylamine $CH_3NH_2$ produces $\cdot CH_2NH_2$, an important radical precursor for the formation of glycine, directly supporting the mechanism, reaction (1), proposed by Ioppolo et al.[19] for the formation of glycine under conditions similar to dark interstellar clouds with no need for UV irradiation or cosmic rays. The mobility of chemical reactants in the bulk ice is assumed through a swapping mechanism that was supported by laboratory work[34,35] and theoretical investigations[5]; this mechanism likely brings H atom and $CH_3NH_2$ in close proximity to react.

## Conclusions

The IR spectrum of $\cdot CH_2NH_2$ is previously unreported; it provides a unique tool to probe this important intermediate, a precursor of glycine. We showed also the order of the consecutive H abstraction of $CH_3NH_2$ to be on the $CH_3$ moiety first (to produce $\cdot CH_2NH_2$), followed by the $NH_2$ moiety (to produce $CH_2NH$); the presence of the back reaction (H addition to radicals) was confirmed in experiments of H + $CD_3NH_2$ to produce $CD_2HNH_2$. This dual-cycle mechanism provides an explanation that $CH_2NH$ and $CH_3NH_2$ might be chemically connected. $CH_2NH$ was also considered to be a precursor of glycine via reactions with CO + $H_2O$ or $CO_2$ + $H_2$ in hot core or cold molecular clouds[36–38].

## Methods

**Experimental details**. A description on the IR absorption matrix-isolation system using $p$-$H_2$ as a matrix host is available elsewhere[22,39,40]. A nickel-coated copper plate at 3.2 K served as a cold substrate for matrix samples and also for reflection absorption spectra. A closed-cycle helium refrigerator system was used to cool the substrate. A gaseous mixture of $CH_3NH_2/Cl_2/p$-$H_2$ (1/10/10000) was deposited, typically over a period of 9 h at a flow rate ~7 STP cm$^3$ min$^{-1}$ (STP indicates standard temperature 273 K and pressure 760 torr). The photolysis of $Cl_2$ in the matrix at 365 nm (light-emitting diode, 5 W) produced Cl atoms; subsequent IR irradiation (unfiltered external SiC source) promoted the reaction Cl + $H_2$ ($v = 1$) → H + HCl, resulting in the generation of H atoms. The matrix was maintained in darkness for 10 h to study H-tunneling reactions and was further photolyzed with light at 460 nm to distinguish lines of each species. To generate $p$-$H_2$, normal $H_2$ (99.9999%) was passed through a trap at 77 K before entering a converter containing an iron(III)-oxide catalyst cooled to 12.9 K with a closed-cycle helium refrigerator. Methylamine ($CH_3NH_2$, Sigma-Aldrich, purity ≥99%), having vapor pressure ~1400 Torr at 293 K, was used to prepare a gaseous mixture $CH_3NH_2/p$-$H_2$ (1/10000). Methylamine-$d_3$ ($CD_3NH_2$, Cambridge Isotope Laboratories, ≥98% isotopic purity) was used for isotopic experiments. A Fourier-transform infrared (FTIR) spectrometer equipped with KBr beam splitter and HgCdTe detector at 77 K was used to record IR absorption spectra covering a spectral range 600−4000 cm$^{-1}$. A total of 200 interferometric scans at a resolution of 0.25 cm$^{-1}$ were typically recorded at each stage of the experiment.

**Quantum-chemical calculations**. Geometry optimizations and vibrational analyses (wavenumbers and IR intensities) were calculated with the Gaussian16 program package[41]. Density-functional theory calculations were performed using B3LYP functionals[42] and standard Dunning's correlation-consistent basis set augmented with diffuse functions, aug-cc-pVTZ[43]. Furthermore, calculations on single-point electronic energies with the method coupled cluster with single and double and perturbative triple excitations, CCSD(T)[44], were performed on geometries obtained with the B3LYP/aug-cc-pVTZ method; zero-point vibrational energies (ZPVE) were corrected according to the harmonic vibrational wavenumbers calculated with the B3LYP method. Both harmonic and anharmonic vibrational wavenumbers were calculated with the B3LYP/aug-cc-pVTZ method. To obtain scaled harmonic vibrational wavenumbers, plots of observed wavenumbers against calculated harmonic vibrational wavenumbers were employed for two separate regions. Two linear equations, $y = (0.9810 \pm 0.0126)\,x − (2.9 \pm 16.9)$ and $y = (0.8907 \pm 0.0084)\,x + (233.0 \pm 27.2)$, were derived for regions 800−1700 and 2900−3600 cm$^{-1}$, respectively; $y$ is the observed wavenumber and $x$ is the calculated harmonic vibrational wavenumber. The average absolute deviation is $5.1 \pm 3.8$ cm$^{-1}$ between experiments and scaled harmonic vibrational wavenumbers of $CH_3NH_2$ and $16.4 \pm 22.7$ cm$^{-1}$ between experiments and anharmonic vibrational wavenumbers of $CH_3NH_2$. The same equations were employed to scale harmonic vibrational wavenumbers of all species considered in this work.

## Data availability

The data that support the plots within this paper and other findings of this study are available from the corresponding author upon reasonable request.

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

## Acknowledgements

This work was supported by Ministry of Science and Technology, Taiwan (Grants nos. MOST 110-2639-M-A49-001-ASP and 110-2634-F-009-026) and the Center for Emergent Functional Matter Science of National Chiao Tung University from The Featured Areas Research Center Program within the framework of the Higher Education Sprout Project by the Ministry of Education (MOE) in Taiwan. The National Center for High-Performance Computation provided computer time.

## Author contributions

P.R.J. performed all computations, experiments, and initial analysis and wrote an initial draft; Y.-P.L. formulated and administered the research project, acquired the funding, finalized the analysis, and wrote the manuscript with contributions from P.R.J.

## Competing interests
