## [Peer Review File · Communications Chemistry]

Reviewers' comments:

Reviewer #1 (Remarks to the Author):

The article « A chemical link between methylamine and methylene imine: identification of aminomethyl radical and implications for interstellar glycine formation" proposed by Prasad Ramesh Joshi¹ and Yuan-Pern Lee deals with the identification of radical species involved in the formation of one of the most important prebiotic molecule (glycine).

The authors used solid p-H₂ matrix IR spectroscopy to perform H + CH₃NH₂ reactions and identify the formation of •CH₂NH₂ and CH₂NH through their vibrational spectra. The formation of •CH₂NH₂ is observed in activation-less environment, which supports its formation in dark interstellar clouds. A multiple step cycle mechanism is given that explains the kinetics and stability of the species. These findings are supporting the hypothesis by Ioppolo et al for the formation of glycine with irradiation involved.

The topic is of great interest and I find the methodology relevant. The article is globally well presented and it might be of interest for Commschem. However I have a few comments that the authors should take into account.

The article relies on the comparison between measured IR spectra and DFT calculations. In order to identify which species is present, the authors compute the spectrum of various possible molecules.

The comparison itself seems to rely on "visual inspection": lines are close in frequency and intensities etc...

I found this methodology rather unreliable, as the "agreement" is subjective. I would like to encourage the authors to define more "objective" criteria (percentage of agreement or a match score) that would define more precisely allow to compare the agreement between the experimental spectrum and theoretical spectra.

The way the data are represented makes the comparison difficult to read (see for instance Figure1). For instance the authors claim:

« The lines in group A at 3500.5, 3403.6, 3143.3, 3042.6, 1609.9, 1213.6, and 685.5 cm⁻¹ 106 107 agree well, in terms of wavenumbers and relative intensities »

"Agree well" does not have any significance, the relative intensity and frequency match is hard to quantify by a simple visual inspection of this figure.

I therefore recommend the author to use a different representation where the spectra are zoomed in and where the comparison exp/theo is direct (superimpose experimental spectra and calculated sticks).

Why calculated lines at 1300 and 1450 cm⁻¹ are not present in experimental data?

overall the work is interesting but it requires a better presentation of the results and more quantitative criteria for comparison.

Reviewer #2 (Remarks to the Author):

The work presented in the manuscript is the follow-up of previous similar investigations on other systems. As in those cases, I have found the present study very interesting and extremely relevant to the cold chemistry of interstellar objects.

In particular, the authors have investigated in solid p-H₂ the effects caused by exposing methylamine to hydrogen atoms. An interesting isotopic effect has also been noted when using partially deuterated methylamine. The employed experimental technique is state-of-the-art and the results are interpreted in the light of dedicated electronic structure calculations.

I have only a few comments that the authors might address.

1) There has been a recent theoretical investigation on the reaction CH₃ + NH₂ assisted by a cluster of 18- or 33-water molecules to simulate the effect of amorphous ice. I think it would be nice to compare part of the present theoretical results with those reported by Enrique-Romero et al. as the real system we aim to understand is not only cold but features the presence of water molecules. The paper is in press in ApJSS, but can be found also here <https://arxiv.org/pdf/2201.10864.pdf>

2) I think that the astrophysical implications of this study have been stretched a little bit too far. I refer to the relation between methanimine and methylamine. First of all, methanimine is almost ubiquitous and has been observed in many different kinds of interstellar objects. Its first detection dates back to 1973. On the contrary, methylamine has been detected in few objects. In other words, there are many environments where methanimine is observed and methylamine is not. This is inconsistent with the mother-daughter relation that is suggested here. Not only that: there are numerous hints that methanimine can be considered a signpost for chemistry occurring in the gas phase, while there is no gaseous route of methylamine formation, see, for instance, Suzuki et al. (ApJ 825, 79, 2016; <https://doi.org/10.3847/0004-637X/825/1/79>).

3) I think that there is an important previous study that should be cited here, that is the study on hydrogenation of solid hydrogen cyanide HCN and methanimine CH₂NH at low temperature by Theule and coworkers (A&A 534, A64, 2011; <https://doi.org/10.1051/0004-6361/201117494>). Interestingly, in that study, the authors reported they have been unable to observe methanimine formation in the hydrogenation of HCN because it easily converts into methylamine. Apparently, they were unable to see the same effect caused by H-abstraction that has been seen here. A comment is in order.

Reviewer #3 (Remarks to the Author):

Manuscript number: COMMSCHEM-22-0003-T

Paper title: A chemical link between methylamine and methylene imine: identification of aminomethyl radical and implications for interstellar glycine formation

Authors: Prasad Ramesh Joshi and Yuan-Pern Lee

The authors report on identification of aminomethyl radical (CH₂NH₂) in solid p-H₂ matrix. The authors combined IR measurements of both normal and partially deuterated CH₂NH₂ with quantum chemical calculations. The impressive

results have a potential to accelerate our understanding of glycine formation in the interstellar molecular clouds. The authors also claim that there is a linear correlation between column densities of CH_3NH_2 and CH_2NH in the interstellar molecular clouds.

However, I found major issues in the paper, described below. I would like the authors to reconsider and to fix these issues. I would like to review the revised manuscript again.

Major Comments:

1. The experimental conditions may not be suitable for comparing with the interstellar conditions.

I have found that the experiments were designed very well. It is a good step to utilize partially deuterated CH_3NH_2 towards secure identification of the CH_2NH radical. The experiments were conducted in the solid H_2 matrix at 3.2 K. This environment is very different from that in the actual physical conditions in the interstellar dust. It is widely understood in astrophysics that the surface of interstellar dust particles, especially deep inside dense molecular clouds, such as hot cores, are covered with amorphous water ice (ice mantle). Further it is also thought in astrophysics that the dust temperature is determined from the balance of their radiative heating (by absorbing far-IR interstellar radiation penetrating into the cores) and cooling (via the continuum emission). Theoretical consideration with these assumptions concluded that the dust temperature will not be lower than 8 K.

I wonder if the beautiful experimental results by the authors could be applied to actual interstellar dust particles. The temperature, 3.2 K, is much lower than the theoretical lower limit of dust temperature. Further, interaction between solid H_2 and CH_2NH would be different from those between water ice and CH_2NH . If the authors claim that it is possible, I would ask the authors to demonstrate the possibility with convincing rationales.

2. Applicability of the quantum chemical calculations to species in solid phase

I fully understand that the quantum chemical calculations are powerful tools for understanding energies of reactants, products as well as transition states. Quantum chemical calculations are often used to predict IR spectra in the "gas phase". The experiments are conducted in the solid H_2 matrix, not in the gas phase. Since I was not able to find a text that the quantum chemical calculations were made for the solid phase, I wonder if the quantum chemical calculations shown in the paper considered that the reactants, products and transition state species are in the solid phase. It seems to me that the calculations correspond to the gas phase. If the quantum chemical calculations corresponded to the gas phase, the authors should not compare the calculated IR spectra and the experimental ones. If the authors made the calculations for the solid phase, I would ask the authors to add new text showing that the quantum chemical calculations are for the solid phase.

3. Dual-cycle mechanism

The authors propose the "dual-cycle mechanism" in Figure 4. However, the right-hand part of the figure has already been pointed out by Garrod (2013; reference 5) and Suzuki et al. (2018; reference 6). Therefore I would suggest the authors to discard Figure 4. If the authors think this figure is valid, I would like the authors to add a new text that similar considerations were made in the past by referring the papers above, and to add another text with regard to new insights contained in this

figure.

4. Claimed linear correlation between interstellar CH_3NH_2 and CH_2NH

I would suggest the authors to discard Figure 5 and relevant text.

In radio astronomical observations, astronomers observe atoms and molecules only in the gas-phase. Suzuki et al. (2016, the Astrophysical Journal, 825, id.79) reported that CH_2NH is primarily formed in gas-phase reactions, primarily between CH_3 and NH radicals. Solid-phase CH_2NH is immediately converted into CH_3NH_2 . Thus CH_3NH_2 is formed in the solid-phase, then are evaporated from the dust surfaces. This would mean that the apparent linear correlation has no physical meanings. I would also point out that the compiled observational results were taken by many telescopes with largely different spatial resolutions. A celestial object shows a spatial structure. When a data is taken with a large spatial resolution, it means an average of molecular distribution within that resolution. When astronomers use small (fine) spatial resolution, it is usually expected to have higher column density.

Reviewer:

Prof. Dr. Masatoshi Ohishi,

National Astronomical Observatory of Japan

Response/revisions to the reviewers' comments

We appreciate very much the valuable comments and suggestions from the reviewers; these comments really helped us to improve the manuscript. Below are the detailed responses to the reviewers' comments. The reviewers' comments are in black, our responses are listed in blue color after each comment, and the revised text are highlighted in yellow.

Reviewer #1 (Remarks to the Author):

The article « A chemical link between methylamine and methylene imine: identification of aminomethyl radical and implications for interstellar glycine formation" proposed by Prasad Ramesh Joshi¹ and Yuan-Pern Lee deals with the identification of radical species involved in the formation of one of the most important prebiotic molecule (glycine).

The authors used solid p-H₂ matrix IR spectroscopy to perform H + CH₃NH₂ reactions and identify the formation of •CH₂NH₂ and CH₂NH through their vibrational spectra. The formation of •CH₂NH₂ is observed in activation-less environment, which supports its formation in dark interstellar clouds. A multiple step cycle mechanism is given that explains the kinetics and stability of the species.

These findings are supporting the hypothesis by Ioppolo et al for the formation of glycine with irradiation involved. The topic is of great interest and I find the methodology relevant. The article is globally well presented and it might be of interest for Commschem.

However I have a few comments that the authors should take into account.

The article relies on the comparison between measured IR spectra and DFT calculations. In order to identify which species is present, the authors compute the spectrum of various possible molecules. The comparison itself seems to rely on "visual inspection": lines are close in frequency and intensities etc...

I found this methodology rather unreliable, as the "agreement" is subjective. I would like to encourage the authors to define more "objective" criteria (percentage of agreement or a match score) that would define more precisely allow to compare the agreement between the experimental spectrum and theoretical spectra.

The way the data are represented makes the comparison difficult to read (see for instance Figure 1). For instance the authors claim: « The lines in group A at 3500.5, 3403.6, 3143.3, 3042.6, 1609.9, 1213.6, and 685.5 cm⁻¹ 106 107 agree well, in terms of wavenumbers and relative intensities » "Agree well" does not have any significance, the relative intensity and frequency match is hard to quantify by a simple visual inspection of this figure. I therefore recommend the author to use a different representation where the spectra are zoomed in and where the comparison exp/theo is direct (superimpose experimental spectra and calculated sticks). Why calculated lines at 1300 and 1450 cm⁻¹ are not present in experimental data?

Overall the work is interesting but it requires a better presentation of the results and more quantitative criteria for comparison.

Response/Revision: Because theoretical predictions typically have errors about $\pm 20 \text{ cm}^{-1}$ and intensities $\pm 50 \%$ or even larger, it is not easy to have superimposed spectra with zoomed scale. Because of this limitation, usually a pattern recognition (wavenumbers and relative intensities) was employed to aid in making the assignments. We agree with the

reviewer that the description was subjective, so in Table 1 we listed the observed and predicted wavenumbers and intensities for comparison. Now we have included information on deviations between experiment and predicted vibrational wavenumbers for all observed wavenumbers in Table 1 and included the average absolute deviations in the text.

As far as predicted lines at 1300 and 1450 cm^{-1} are concerned, the intensity of both lines is 2 km mol^{-1} , which is $\sim 1\%$ of the most intense line, and thus these lines are difficult to observe experimentally.

We added two sentences on page 6, which reads: “The average absolute deviation between experiment and prediction is $18.7 \pm 15.2 \text{ cm}^{-1}$ ($1.06 \pm 0.8\%$) for $\bullet\text{CH}_2\text{NH}_2$. The large deviation for ν_6 (CH_2 wag) is typical for this mode because of the inadequacy in describing the double-well potential experienced by N atom, similar to NH_3 . All lines of $\bullet\text{CH}_2\text{NH}_2$ located in our detection spectral range with predicted IR intensity greater than 6 km mol^{-1} were observed; predicted lines near 1448 and 1292 cm^{-1} have intensity $\sim 2 \text{ km mol}^{-1}$ too small to be observed.” We also added one column in Table 1 to show the deviations between experiments and predictions.

Reviewer #2 (Remarks to the Author):

The work presented in the manuscript is the follow-up of previous similar investigations on other systems. As in those cases, I have found the present study very interesting and extremely relevant to the cold chemistry of interstellar objects. In particular, the authors have investigated in solid $p\text{-H}_2$ the effects caused by exposing methylamine to hydrogen atoms. An interesting isotopic effect has also been noted when using partially deuterated methylamine. The employed experimental technique is state-of-the-art and the results are interpreted in the light of dedicated electronic structure calculations.

I have only a few comments that the authors might address.

1) There has been a recent theoretical investigation on the reaction $\text{CH}_3 + \text{NH}_2$ assisted by a cluster of 18- or 33-water molecules to simulate the effect of amorphous ice. I think it would be nice to compare part of the present theoretical results with those reported by Enrique-Romero et al. as the real system we aim to understand is not only cold but features the presence of water molecules. The paper is in press in *ApJSS*, but can be found also here <https://arxiv.org/pdf/2201.10864.pdf>

Response/Revision: We thank the reviewer for providing this information. Following this idea to understand the influence of surrounding H_2 on $\bullet\text{CH}_2\text{NH}_2$ in solid $p\text{-H}_2$, we performed quantum-chemical calculations with eighteen H_2 molecules surrounding $\bullet\text{CH}_2\text{NH}_2$; the H_2 molecules were placed either in a hexagonal-closed pack (*hcp*) lattice or randomly (i.e. freely optimized). As expected, the simulated IR stick spectra in both solid-phase cases are similar to the gaseous phase, within uncertainties of calculations. We added the following text on page 5.

“To understand the perturbations of H_2 on the IR spectrum of $\bullet\text{CH}_2\text{NH}_2$, we performed calculations also on $\bullet\text{CH}_2\text{NH}_2$ surrounded by eighteen H_2 molecules, either in a hexagonal-closed pack (*hcp*) lattice or randomly (free optimization). The resultant vibrational wavenumbers and IR intensities are compared in Supplementary Table 1 and the simulated IR stick spectra of $\bullet\text{CH}_2\text{NH}_2$ are presented in Supplementary Fig. 2 to compare with calculations for gaseous $\bullet\text{CH}_2\text{NH}_2$ and experiments. The perturbation by

H₂ is small (with average absolute deviations 8.8 ± 6.0 and 14.8 ± 8.5 cm⁻¹ from the gaseous phase; listed errors represent one standard deviation in fitting) and within calculation errors. This is in line with the fact that observed IR spectra of matrix-isolated species typically showed <1 % matrix shifts so that comparison of observed vibrational wavenumbers with predictions of gaseous species was generally performed.”

The suggested reference simulated the stability and reactivity of species on ice surface, which is different from our calculations, which simulated the spectrum of the radicals. To include this reference, we added a few sentences on page 11 after Fig. 4 as: “We understand that our experimental conditions do not mimic the ISM conditions closely, so our results cannot be applied directly to the reactions in the ISM. For example, in the case of water ice environments, the interaction between water and the guest species might be stronger so that the stability and reactivity of radicals are different from the gaseous phase, as demonstrated by the simulations of radical-radical reactions on icy surfaces by Enrique-Romero *et al.*³³”.

2) I think that the astrophysical implications of this study have been stretched a little bit too far. I refer to the relation between methanimine and methylamine. First of all, methanimine is almost ubiquitous and has been observed in many different kinds of interstellar objects. Its first detection dates back to 1973. On the contrary, methylamine has been detected in few objects. In other words, there are many environments where methanimine is observed and methylamine is not. This is inconsistent with the mother-daughter relation that is suggested here. Not only that: there are numerous hints that methanimine can be considered a signpost for chemistry occurring in the gas phase, while there no gaseous route of methylamine formation, see, for instance, Suzuki *et al.* (ApJ 825, 79, 2016; <https://doi.org/10.3847/0004-637X/825/1/79>).

Response/Revision: We thank the reviewer for point this out. We have removed Figure 5 and the associated text discussing interstellar [CH₂NH]/[CH₃NH₂] ratio from the manuscript.

3) I think that there is an important previous study that should be cited here, that is the study on hydrogenation of solid hydrogen cyanide HCN and methanimine CH₂NH at low temperature by Theule and coworkers (A&A 534, A64, 2011; <https://doi.org/10.1051/0004-6361/201117494>). Interestingly, in that study, the authors reported they have been unable to observe methanimine formation in the hydrogenation of HCN because it easily converts into methylamine. Apparently, they were unable to see the same effect caused by H-abstraction that has been seen here. A comment is in order.

Response/Revision: We have cited the recommended article and added a new text which reads: “The first H-abstraction channel (reaction 1) depicted in Fig. 4 was reported by Garrod⁵ and Suzuki⁶ in their theoretical models. Hydrogenation of solid HCN and CH₂NH at low temperature conducted by Theule *et al.*³² resulted in the formation of CH₃NH₂ directly, indicating the presence of two consecutive H addition (reactions 3 and 4), even though •CH₂NH₂ was not observed directly. Our observations of the formation of •CH₂NH₂ and CH₂NH in darkness and the formation of CD₂HNH₂ from H reactions with CD₃NH₂ further support the dual-cycle mechanism.”

Reviewer #3 (Remarks to the Author):

Manuscript number: COMMSCHEM-22-0003-T

Paper title: A chemical link between methylamine and methylene imine: identification of aminomethyl radical and implications for interstellar glycine formation

Authors: Prasad Ramesh Joshi and Yuan-Pern Lee

The authors report on identification of aminomethyl radical (CH_2NH_2) in solid *p*- H_2 matrix. The authors combined IR measurements of both normal and partially deuterated CH_2NH_2 with quantum chemical calculations. The impressive results have a potential to accelerate our understanding of glycine formation in the interstellar molecular clouds. The authors also claim that there is a linear correlation between column densities of CH_3NH_2 and CH_2NH in the interstellar molecular clouds.

However, I found major issues in the paper, described below. I would like the authors to reconsider and to fix these issues. I would like to review the revised manuscript again.

Major Comments:

1. The experimental conditions may not be suitable for comparing with the interstellar conditions.

I have found that the experiments were designed very well. It is a good step to utilize partially deuterated CH_3NH_2 towards secure identification of the CH_2NH_2 radical. The experiments were conducted in the solid *p*- H_2 matrix at 3.2 K. This environment is very different from that in the actual physical conditions in the interstellar dust. It is widely understood in astrophysics that the surface of interstellar dust particles, especially deep inside dense molecular clouds, such as hot cores, are covered with amorphous water ice (ice mantle). Further it is also thought in astrophysics that the dust temperature is determined from the balance of their radiative heating (by absorbing far-IR interstellar radiation penetrating into the cores) and cooling (via the continuum emission). Theoretical consideration with these assumptions concluded that the dust temperature will not be lower than 8 K.

I wonder if the beautiful experimental results by the authors could be applied to actual interstellar dust particles. The temperature, 3.2 K, is much lower than the theoretical lower limit of dust temperature. Further, interaction between solid H_2 and CH_2NH_2 would be different from those between water ice and CH_2NH_2 . If the authors claim that it is possible, I would ask the authors to demonstrate the possibility with convincing rationales.

Response/Revision: We thank the reviewer for pointing this out. Firstly, we would like to emphasize that the main goal of the present study is to demonstrate the formation of $\bullet\text{CH}_2\text{NH}_2$ radical which was considered to be an intermediate for the reaction $\text{CH}_3\text{NH}_2 + \text{HOCO}$ leading to the formation of interstellar glycine. We do not claim that our experimental conditions mimic the ISM conditions, rather, we would like to explore key reactions and radical intermediates that are difficult to study under interstellar relevant conditions because of the complicated interactions with species in ice. The temperature used for our experiments is lower than the theoretically assumed dust temperature of 8 K because solid *p*- H_2 matrix evaporated at 5 K. However, typically, if the reaction occurs at

3.2 K, it is expected to occur also at 8 K. Secondly, we have presented that H abstraction of CH_3NH_2 is significant for the formation of $\bullet\text{CH}_2\text{NH}_2$. Earlier laboratory (e.g. Ioppolo, S. et al. *Nat. Astron.* **5**, 197–205 (2021) and theoretical (e.g. Garrod, R. T. *Astrophys. J.* **765**, 1–29 (2013)) models assumed the formation of $\bullet\text{CH}_2\text{NH}_2$ via H abstraction of CH_3NH_2 by H atoms or $\bullet\text{OH}$. Three-phase model demonstrated by Garrod included H-abstraction reactions taking place between complex molecules (precursors) and the most significant grain-surface radicals/atoms such as H, $\bullet\text{OH}$, and $\bullet\text{NH}_2$. Additionally, the mobility of chemical reactants in the bulk ice is assumed through a swapping mechanism that was supported by laboratory work (e.g. Oberg et al. *Astron. Astrophys.* **505**, 183-194 (2009). And Feyolle et al. *Astron. Astrophys.* **529**, A74 (2011) and theoretical investigations (e.g. Garrod, R. T. *Astrophys. J.* **765**, 1–29 (2013)); this mechanism likely brings H atom and CH_3NH_2 in close proximity to react.

To clarify this, we have revised the paragraph after Fig. 4 (pages 11 and 12) as: “We understand that our experimental conditions do not mimic the ISM conditions closely, so our results cannot be applied directly to the reactions in the ISM. For example, in the case of water ice environments, the interaction between water and the guest species might be stronger so that the stability and reactivity of radicals are different from the gaseous phase, as demonstrated by the simulations of radical-radical reactions on icy surfaces by Enrique-Romero *et al.*³³ Nevertheless, our results clearly indicate that reaction of H with methylamine CH_3NH_2 produces $\bullet\text{CH}_2\text{NH}_2$, an important radical precursor for the formation of glycine, directly supporting the mechanism, reaction (1), proposed by Ioppolo et al.¹⁹ for the formation of glycine under conditions similar to dark interstellar clouds with no need for UV irradiation or cosmic rays. The mobility of chemical reactants in the bulk ice is assumed through a swapping mechanism that was supported by laboratory work^{34,35} and theoretical investigations;⁵ this mechanism likely brings H atom and CH_3NH_2 in close proximity to react.”

2. Applicability of the quantum chemical calculations to species in solid phase

I fully understand that the quantum chemical calculations are powerful tools for understanding energies of reactants, products as well as transition states. Quantum chemical calculations are often used to predict IR spectra in the “gas phase”. The experiments are conducted in the solid H_2 matrix, not in the gas phase. Since I was not able to find a text that the quantum chemical calculations were made for the solid phase, I wonder if the quantum chemical calculations shown in the paper considered that the reactants, products and transition state species are in the solid phase. It seems to me that the calculations correspond to the gas phase. If the quantum chemical calculations corresponded to the gas phase, the authors should not compare the calculated IR spectra and the experimental ones. If the authors made the calculations for the solid phase, I would ask the authors to add new text showing that the quantum chemical calculations are for the solid phase.

Response/Revision: The nice thing about matrix isolation for IR spectroscopy is that, even it is in the solid phase, the perturbation by the matrix host is small because of negligible interactions between guest and host molecules. The vibrational wavenumbers in solid Ar differed by those in the gaseous phase by less than 1 % (Jacox M.E. Chem.

Phys. **189**, 149–170 (1994)). The perturbation by *p*-H₂ is even smaller than that by Ar. Hence, typically, researchers compared the matrix results with predictions of gaseous species for spectral identification. To address the reviewer's concern, we have performed additional calculations with eighteen H₂ molecules surrounding •CH₂NH₂; the H₂ molecules were placed either in a hexagonal-closed pack (hcp) lattice or randomly (freely optimized). As expected, the simulated IR stick spectra in both solid-phase cases are similar to the gaseous phase. We added the following text on page 5.

“To understand the perturbations of H₂ on the IR spectrum of •CH₂NH₂, we performed calculations also on •CH₂NH₂ surrounded by eighteen H₂ molecules, either in a hexagonal-closed pack (*hcp*) lattice or randomly (free optimization). The resultant vibrational wavenumbers and IR intensities are compared in Supplementary Table 1 and the simulated IR stick spectra of •CH₂NH₂ are presented in Supplementary Fig. 2 to compare with calculations for gaseous •CH₂NH₂ and experiments. The perturbation by H₂ is small (with average absolute deviations 8.8 ± 6.0 and 14.8 ± 8.5 cm⁻¹ from the gaseous phase; listed errors represent one standard deviation in fitting) and within calculation errors. This is in line with the fact that observed IR spectra of matrix-isolated species typically showed <1 % matrix shifts so that comparison of observed vibrational wavenumbers with predictions of gaseous species was generally performed.”

3. Dual-cycle mechanism

The authors propose the “dual-cycle mechanism” in Figure 4. However, the right-hand part of the figure has already been pointed out by Garrod (2013; reference 5) and Suzuki et al. (2018; reference 6). Therefore I would suggest the authors to discard Figure 4. If the authors think this figure is valid, I would like the authors to add a new text that similar considerations were made in the past by referring the papers above, and to add another text with regard to new insights contained in this figure.

Response/Revision: We thank the reviewer for pointing this out. The recommended articles by Garrod (reference 5) and Suzuki et al. (reference 6) provided only the H-abstraction reaction resulting in the formation of •CH₂NH₂ through reaction 1 presented in Fig. 4. Therefore, Fig. 4 is necessary since it provides an overview of the products formed through stepwise H-abstraction and H-addition reactions. Moreover, formation of CD₂HNH₂ in darkness in the H + CD₃NH₂ reaction confirms H-addition reactions. We have revised the paragraph before Fig. 4 on page 11 as “The observation of consecutive H abstraction of CH₃NH₂ to form •CH₂NH₂ and CH₂NH, and their H-addition to reform CH₃NH₂ and •CH₂NH₂, respectively, connects CH₃NH₂, •CH₂NH₂, and CH₂NH via a dual-cycle mechanism shown in Fig. 4, similar to that among formamide HC(O)NH₂, H₂NCO and HNCO.²⁵ The first H-abstraction channel (reaction 1) depicted in Fig. 4 was reported by Garrod⁵ and Suzuki⁶ in their theoretical models. Hydrogenation of solid HCN and CH₂NH at low temperature conducted by Theule et al.³² resulted in the formation of CH₃NH₂ directly, indicating the presence of two consecutive H addition (reactions 3 and 4), even though •CH₂NH₂ was not observed directly. Our observations of the formation of •CH₂NH₂ and CH₂NH in darkness and the formation of CD₂HNH₂ from H reactions with CD₃NH₂ further support the dual-cycle mechanism.”

4. Claimed linear correlation between interstellar CH_3NH_2 and CH_2NH

I would suggest the authors to discard Figure 5 and relevant text.

In radio astronomical observations, astronomers observe atoms and molecules only in the gas-phase. Suzuki et al. (2016, the Astrophysical Journal, 825, id.79) reported that CH_2NH is primarily formed in gas-phase reactions, primarily between CH_3 and NH radicals. Solid-phase CH_2NH is immediately converted into CH_3NH_2 . Thus CH_3NH_2 is formed in the solid-phase, then are evaporated from the dust surfaces. This would mean that the apparent linear correlation has no physical meanings. I would also point out that the compiled observational results were taken by many telescopes with largely different spatial resolutions. A celestial object shows a spatial structure. When a data is taken with a large spatial resolution, it means an average of molecular distribution within that resolution. When astronomers use small (fine) spatial resolution, it is usually expected to have higher column density.

Response/Revision: We thank the reviewer for this valuable comments. We have deleted Fig. 5 and related text from the manuscript.

Reviewers' comments:

Reviewer #1 (Remarks to the Author):

I read the new version of the article.

The authors have answered my questions et included relevant modifications.

I therefore recommend publication.

Reviewer #2 (Remarks to the Author):

The authors have carefully addressed my concerns and I am satisfied with their revision.

In my opinion, the manuscript can be accepted as it is.

Reviewer #3 (Remarks to the Author):

Manuscript number: COMMSCHEM-22-0003-A

Paper title: A chemical link between methylamine and methylene imine: identification of aminomethyl radical and implications for interstellar glycine formation

Authors: Prasad Ramesh Joshi and Yuan-Pern Lee

The authors report on identification of aminomethyl radical (CH_2NH_2) in solid p- H_2 matrix. The authors combined IR measurements of both normal and partially deuterated CH_2NH_2 with quantum chemical calculations. The impressive results have a potential to accelerate our understanding of glycine formation in the interstellar molecular clouds. The authors also claim that there is a linear correlation between column densities of CH_3NH_2 and CH_2NH in the interstellar molecular clouds.

The authors addressed all comments from the reviewers, resulting in improving the content and the quality of the paper.

I still have a few issues in the paper, described below. I would like the authors to consider these issues and to fix them if needed. I would like to review the revised manuscript again for the (possibly) final round.

Comments:

1. Quantum chemical calculations on $\text{CH}_3\text{NH}_2/\text{CH}_2\text{NH}_2/\text{CH}_2\text{NH}$ (Figure 2)

Figure 2 contains quantum chemical calculations to be compared with the experimental results. In the middle column of the figure (light blue color), the initial reactants are a single CH_3NH_2 and four H atoms, not a single H atom. Do you need four H atoms, not a single H atom, to proceed the H-abstraction reactions, despite that the authors describe $\text{H} + \text{CH}_3\text{NH}_2$ in many places of the paper? In Figure 4 (the proposed dual-cycle mechanism) there is only one H atom to react with CH_3NH_2 . It seems to me that Figures 2 and 4 are not consistent. I would like the authors to clarify if four H atoms are needed for the H-abstraction reactions, and add new text to describing that such reactions really occurred in your experiment, if needed.

2. A new discussion related with Theule et al. (2011)

Theule et al. (2011) conducted H additions to HCN, and succeeded to form CH_2NH and CH_3NH_2 . The

formation of CH_3NH_2 from CH_2NH would correspond to the quantum chemical calculations by the authors (most left part of Figure 2) (except for excessive H_2 molecules). As the authors have recognized, Theule et al. did not observe the CH_2NH_2 radical. Would it be possible for the authors to discuss why the authors succeeded to observe the IR spectra of the CH_2NH_2 radical despite Theule et al. did not? Is this because of the surrounding conditions between two experiments, in solid H_2 or pure $\text{HCN}/\text{CH}_2\text{NH}$, or different temperatures? I believe that such a discussion would be useful for researchers in low temperature chemistry and/or interstellar chemistry.

Response/revisions to the reviewers' comments

We appreciate very much the valuable comments from reviewer 3. Below are the detailed responses to the reviewer's comments. The reviewer's comments are in black, our responses are listed in blue color after each comment, and the revised text are highlighted in yellow.

Reviewer #3 (Remarks to the Author):

Manuscript number: COMMSCHEM-22-0003-A

Paper title: A chemical link between methylamine and methylene imine: identification of aminomethyl radical and implications for interstellar glycine formation

Authors: Prasad Ramesh Joshi and Yuan-Pern Lee

The authors report on identification of aminomethyl radical (CH_2NH_2) in solid p-H₂ matrix. The authors combined IR measurements of both normal and partially deuterated CH_2NH_2 with quantum chemical calculations. The impressive results have a potential to accelerate our understanding of glycine formation in the interstellar molecular clouds. The authors also claim that there is a linear correlation between column densities of CH_3NH_2 and CH_2NH in the interstellar molecular clouds.

The authors addressed all comments from the reviewers, resulting in improving the content and the quality of the paper.

I still have a few issues in the paper, described below. I would like the authors to consider these issues and to fix them if needed. I would like to review the revised manuscript again for the (possibly) final round.

Comments:

1. Quantum chemical calculations on $\text{CH}_3\text{NH}_2/\text{CH}_2\text{NH}_2/\text{CH}_2\text{NH}$ (Figure 2)

Figure 2 contains quantum chemical calculations to be compared with the experimental results. In the middle column of the figure (light blue color), the initial reactants are a single CH_3NH_2 and four H atoms, not a single H atom. Do you need four H atoms, not a single H atom, to proceed the H-abstraction reactions, despite that the authors describe $\text{H} + \text{CH}_3\text{NH}_2$ in many places of the paper? In Figure 4 (the proposed dual-cycle mechanism) there is only one H atom to react with CH_3NH_2 . It seems to me that Figures 2 and 4 are not consistent. I would like the authors to clarify if four H atoms are needed for the H-abstraction reactions, and add new text to describing that such reactions really occurred in your experiment, if needed.

Response/Revision: because Figure 2 shows the potential-energy scheme (PES) for the $\text{H} + \text{CH}_3\text{NH}_2$ system including two successive H-abstraction and three H-addition reactions, we need to balance the reactants (hydrogen atoms and molecules) and also the energies. If we start from the bottom of the blue section (CH_3NH_2), we need two H atoms for H addition and two more for H abstraction to reach the top state of CH_3NH_2 ; in the meantime, these four H atoms turn into 2 H_2 . That is why we have $\text{CH}_3\text{NH}_2 + 4\text{H}$ on top, and $\text{CH}_3\text{NH}_2 + \text{H}_2$ at the bottom (with an energy difference representing the energy of $4\text{H} \rightarrow 2\text{H}_2$). To avoid confusion, we revised the sentence as “The potential-energy scheme

for H-abstraction (left side) and H-addition (right side) is shown in Fig. 2; to be self-consistent in terms of energy and species involved, we included all H atoms and H₂ involved in this reaction network.” on page 7.

We also found that we forgot to balance the hydrogen in the fragmentation products. We have now revised Figure 2 to include three H atoms to the H-addition fragmentation products (CH₄ + NH₂ and CH₃ + NH₃) to balance these reactions.

On the other hand, Figure 4 shows the dual-cycle mechanism, in which no energy was involved in the Figure so that individual step can be presented without worrying about the total energy balance.

2. A new discussion related with Theule et al. (2011)

Theule et al. (2011) conducted H additions to HCN, and succeeded to form CH₂NH and CH₃NH₂. The formation of CH₃NH₂ from CH₂NH would correspond to the quantum chemical calculations by the authors (most left part of Figure 2) (except for excessive H₂ molecules). As the authors have recognized, Theule et al. did not observe the CH₂NH₂ radical. Would it be possible for the authors to discuss why the authors succeeded to observe the IR spectra of the CH₂NH₂ radical despite Theule et al. did not? Is this because of the surrounding conditions between two experiments, in solid H₂ or pure HCN/CH₂NH, or different temperatures? I believe that such a discussion would be useful for researchers in low temperature chemistry and/or interstellar chemistry.

Response/Revision: In the present study, we have taken the advantages of *p*-H₂ matrix isolation in producing and trapping free radicals; the species are well isolated and the hydrogenation reaction was carried out within the solid matrix in a small scale, as compared with hydrogen bombardment so that the radicals have much better chance to survive.

In contrast, hydrogen bombardment on pure HCN/CH₂NH ice were employed in the work demonstrated by Theule et al., in which the •CH₂NH₂ generated during H + CH₂NH reaction would have immediately reacted with another H atom to produce CH₃NH₂ through the barrierless reaction. We added the following text on page 11 “In the present study, all guest molecules are well isolated in solid *p*-H₂ at low temperature so that free radicals such as •CH₂NH₂ have much better chance to be trapped and maintained. Furthermore, because the hydrogenation experiments by Theule et al.³² were carried out by hydrogen bombardment, •CH₂NH₂ is expected to react readily with a second hydrogen atom to form the end product CH₃NH₂.”

REVIEWERS' COMMENTS:

Reviewer #3 (Remarks to the Author):

Manuscript number: COMMSCHEM-22-0003-B

Paper title: A chemical link between methylamine and methylene imine: identification of aminomethyl radical and implications for interstellar glycine formation

Authors: Prasad Ramesh Joshi and Yuan-Pern Lee

The authors appropriately addressed additional comments from the reviewer. The newly added text to explain the difference between the authors' study and Theule et al. sounds reasonable to me.

Now I am happy to endorse the publication of this paper to the Nature Chemistry.